# Whole-Genome Transcript Expression Profiling Reveals Novel Insights into Transposon Genes and Non-Coding RNAs during Atlantic Salmon Seawater Adaptation

**DOI:** 10.3390/biology11010001

**Published:** 2021-12-21

**Authors:** Valentina Valenzuela-Muñoz, Cristian Gallardo-Escárate, Bárbara P. Benavente, Diego Valenzuela-Miranda, Gustavo Núñez-Acuña, Hugo Escobar-Sepulveda, Juan Antonio Váldes

**Affiliations:** 1Interdisciplinary Center for Aquaculture Research (INCAR), University of Concepción, Concepcion 4030000, Chile; crisgallardo@udec.cl (C.G.-E.); Bbenavente@udec.cl (B.P.B.); divalenzuela@udec.cl (D.V.-M.); gustavonunez@udec.cl (G.N.-A.); hescobar.sepulveda@gmail.com (H.E.-S.); jvaldes@unab.cl (J.A.V.); 2Laboratorio de Biotecnología Molecular, Facultad de Ciencias de la Vida, Universidad Andrés Bello, Santiago 8370035, Chile; 3Laboratory of Biotechnology and Aquatic Genomics, Department of Oceanography, University of Concepción, Concepcion 4030000, Chile

**Keywords:** Atlantic salmon, smoltification, genome, mRNAs, miRNAs

## Abstract

**Simple Summary:**

This study proposes a novel approach to analyze transcriptome data sets using the Atlantic salmon seawater adaptation process as a model. *Salmon salar* smolts were transferred to seawater under two strategies: (i) fish group exposed to gradual salinity changes (GSC) and (ii) fish group exposed to a salinity shock (SS). mRNA and miRNAs sequencing were performed for gills, intestine, and head kidney tissues. The whole-genome transcript expression profiling revealed specific gene expression patterns among the tissues and treatments. A great abundance of transposable elements was observed in chromosome regions differentially expressed under experimental conditions. Moreover, small RNA expression analysis suggested fewer of miRNAs associated with the smoltification process. However, target analysis of these miRNAs suggests a regulatory role of process such as growth, stress response, and immunity. The findings uncover whole-transcriptome modulation during seawater adaptation of Atlantic salmon, evidencing the interplaying among mRNAs and miRNAs.

**Abstract:**

The growing amount of genome information and transcriptomes data available allows for a better understanding of biological processes. However, analysis of complex transcriptomic experimental designs involving different conditions, tissues, or times is relevant. This study proposes a novel approach to analyze complex data sets combining transcriptomes and miRNAs at the chromosome-level genome. Atlantic salmon smolts were transferred to seawater under two strategies: (i) fish group exposed to gradual salinity changes (GSC) and (ii) fish group exposed to a salinity shock (SS). Gills, intestine, and head kidney samples were used for total RNA extraction, followed by mRNA and small RNA illumina sequencing. Different expression patterns among the tissues and treatments were observed through a whole-genome transcriptomic approach. Chromosome regions highly expressed between experimental conditions included a great abundance of transposable elements. In addition, differential expression analysis showed a greater number of transcripts modulated in response to SS in gills and head kidney. miRNA expression analysis suggested a small number of miRNAs involved in the smoltification process. However, target analysis of these miRNAs showed a regulatory role in growth, stress response, and immunity. This study is the first to evidence the interplaying among mRNAs and miRNAs and the structural relationship at the genome level during Atlantic salmon smoltification.

## 1. Introduction

Genomics tools have facilitated the elucidation of the global genomic changes under different conditions. Nowadays, the availability of the Atlantic salmon (*Salmo salar*) genome [1] allows for the identification of genome regions associated with pivotal biological processes and responses to the aquatic environmental. One of the most important biological processes during the salmon lifecycle is the parr-smolt transformation (smoltification), which is primarily influenced by water temperature and photoperiod [1]. This process involves physiological, morphological, endocrinal, and behavioral changes [2,3], which have been extensively studied, owing to their implications in salmon aquaculture [1,4,5,6].

Previous transcriptomic analyses have elucidated expression changes of genes associated with growth, metabolism, oxygen transport, osmoregulation, protein biosynthesis, and sensory reception during the Atlantic salmon smoltification process [7,8]. Recently, the role of non-coding RNAs (ncRNAs) during Atlantic salmon transition from freshwater (FW) to seawater (SW) has been proposed as a novel regulatory molecular mechanism involved in fish biology. For instance, previous studies have reported 2864 long non-coding RNAs (lncRNAs) differentially modulated in Atlantic salmon gills during the transition from FW to SW [9]. Among them, two putative lncRNAs with the potential to be used as smoltification-timing biomarkers were identified. One was highly regulated in FW, and the other was upregulated in SW. In addition, a putative regulation role of lncRNAs associated with Na^+^/K^+^-ATPase genes, hormone receptors, and thyroid hormone receptors was suggested [9].

Among ncRNAs, microRNAs (miRNAs) are crucial in post-transcriptional regulation, binding to target mRNAs in 3′UTR and repressing translation to proteins [10,11]. Different biological process, such as development, growth, cell division, metabolism, and apoptosis, are regulated by miRNAs [12,13,14,15]. For Atlantic salmon, 472 miRNAs have been reported in the miRBase [16]. In addition, a total of 521 miRNAs have been described for Atlantic salmon, with different expression patterns among kidney, head kidney, heart, brain, gills, white muscle, and intestine tissues [17]. Moreover, 71 miRNAs were reported to be differently modulated in head kidney of Atlantic salmon during the smoltification process [18]. Furthermore, the authors reported a negative-correlation expression pattern of miRNAs with genes associated with hormone biosynthesis, stress management, immune response, and ion transport. In addition, the study reported a cluster of 37 miRNAs highly regulated before the smoltification initiation and another cluster of 17 miRNAs that increased their expression until SW transfer [18]. Despite the evidence suggesting a role of ncRNAs in the regulation of the smoltification process, comprehensive transcriptome analyses linking physiological adaptation to seawater with the complexity of the salmon genome are unexplored. For instance, there is no evidence of how unduplicated and/or duplicated genes involved in the smoltification are modulated and which are the key molecular elements involved. Herein, the molecular interplaying among coding/non-coding genes expressed in different tissues during seawater adaptation is uncovered. The major obstacle to conducting extensive experimental trials is combining the transcriptome time-series with different sequencing approaches (e.g., mRNA vs. small RNA sequencing), and joining the analyzed target-tissues with the experimental conditions tested. The massive amount of transcriptome data is frequently completely unused or at least not straightforwardly used to identify the primary biological processes modulated and their molecular components. Furthermore, evidence that chromatin in the interphase nucleus has nonrandom localization supports the idea that the nuclear architecture is comprised of three-dimensional (3D) genome spaces. This spatial organization plays crucial roles in genome function and cellular processes, such as DNA replication [19,20], transcription [21], DNA-damage repair [22,23], development, and cell differentiation [24]. Chromosomes occupy distinct subnuclear territories, with transcriptionally active loci positioned at their surface [25,26,27,28,29,30]. Thus, the relationship between gene transcription, gene regulation, and spatial 3D genome structure for relevant biological processes requires further scientific investigation. This study aimed to explore global transcriptome modulation during Atlantic salmon seawater adaptation. Herein, a novel approach was developed to analyze time-series differential transcription data from head kidney, gills, and intestine tissues exposed to salinity changes. In parallel, transcriptional dynamics were associated with chromosome regions where differently expressed thresholds between salinity conditions were identified. Notably, transcriptome analyses revealed novel insights into transposon genes and non-coding RNAs involved in smoltification in Atlantic salmon. This study is the first to suggest putative chromosome regions transcriptionally activated in response to salinity stress in anadromous fish. 

## 2. Materials and Methods

### 2.1. Smoltification and Seawater Transfer

Atlantic salmon smolts (60 ± 6.2 gr) were obtained from a commercial farm (Hendrix Genetics, Boxmeer, The Netherlands) and then transported to the Marine Biology Station, Universidad de Concepción, Dichato, Chile. Fish were maintained in ultraviolet-treated saltwater by single-pass flow-through tank systems on a 12:12 h light: dark cycle, fed daily with a commercial diet, dissolved oxygen level of 8.5 mg/L and pH = 8.0. After ten days of acclimation, a group of 30 smolts was exposed to a gradual salinity change (GSC) by increasing FW salinity to SW. The gradient was set at three salinity points, changing 10 PSU per week over a month. Meanwhile, another group of 30 smolts was exposed to a salinity shock (SS), directly from FW to 32 PSU. Gills, head kidney, and intestine samples were collected during the experiment trial. Samples were collected at FW, 10, 20, and 32 PSU for the GSC group and after a week of acclimation at 32 PSU for the SS group (Appendix A). Both processes were conducted in triplicate. The samples were fixed in RNAlater^®^ (Thermofisher, Waltham, MA, USA) for subsequent RNA isolation. Furthermore, the salmon condition to SW transfer was evaluated by immune histochemistry analyses performed by the VEHICE company. In addition, RT-qPCR expression analysis of ATPase-α and ATPase-β was determined using comparative ΔΔCt relative expression analysis. Primers and qPCR conditions were similar, as described by Valenzuela-Muñoz, Váldes, and Gallardo-Escárate [9]. All animal procedures were carried out under the guidelines approved by the Ethics Committee of the University of Concepción. The experimental design for the current study considered the Three Rs (3Rs) guidelines for animal testing.

### 2.2. High-Throughput Transcriptome Sequencing

Total RNA was isolated from each experimental fish group using TRizol Reagent (Ambion^®^, Austin, TX, USA), following the manufacturer’s instructions. The isolated RNA was evaluated by TapeStation 2200 (Agilent Technologies Inc., Santa Clara, CA, USA), using the R6K Reagent Kit. Three biological replicates were separately sequenced by tissue and sampling point from each experimental fish group. For each replicate, five individuals were used for the RNA extraction and then pooled for library preparation. RNAs with RIN > 8.0 were used for double-stranded cDNA library construction using the TruSeq RNA Sample Preparation Kit v2 (Illumina^®^, San Diego, CA, USA). The same RNA samples were used for small RNA library synthesis using the TruSeq Small RNA Library Prep Kit (Illumina^®^, San Diego, CA, USA). All libraries made for RNA and small RNAs were sequenced by the Hiseq (Illumina^®^, San Diego, CA, USA) platform in Macrogen Inc. Raw data used for the current study are available in SRA-NCBI (Bioproject # PRJNA761374). 

### 2.3. Whole-Genome Transcript Expression Analysis

Raw data from each experimental group were separately trimmed and mapped to the Atlantic salmon genome (GCF_000233385.1) using CLC Genomics Workbench v21 software (Qiagen Bioinformatics, Hilden, Germany). Threshold values for mRNA and small RNA were calculated from the coverage analysis using the Graph Threshold Areas tool in CLC Genomics Workbench v21 software. Here, an index denoted as chromosome genome expression (CGE) was formulated to explore the whole-genome transcript expression profiling according to: $$CGE=\frac{\left|{Χ}_{1}-{Χ}_{n}\right|\;\times \;100}{\left({Χ}_{1}-{Χ}_{n}\right)}$$
where X corresponds to mean coverage of transcripts mapped into a specific chromosome region and compared among experimental conditions (e.g., GSC cv. SS; tissues vs. experimental time-points, mRNAs vs. miRNAs). The transcript coverage values for each dataset were calculated using a threshold of 10,000 to 90,000 reads, where a window size of 10 positions was set to calculate and identify differentially transcribed chromosome regions. The CGE index represents the percentage of transcriptional variation between two or more groups for the same locus. This approach allows for visualization of actively transcribed chromosome regions, identification of differentially expressed loci, exploration of mRNA-miRNAs interactions in term of transcriptional activity, and observation of tissue-specific patterns in fish exposed to several experimental conditions. Finally, threshold values for each dataset and CGE index were visualized in Circos plots [31].

### 2.4. RNA-Seq Data Analysis

Raw sequencing reads were assembled de novo separately for each tissue using the CLC Genomics Workbench v21 software (CLC Bio, Aarhus, Denmark). Assembly was performed with overlap criteria of 70% and a similarity of 0.9 to exclude paralogous sequence variants (Renaut et al., 2010). The settings used were set as mismatch cost = 2, deletion cost = 3, insert cost = 3, minimum contig length = 200 base pairs, and trimming quality score = 0.05. After assembly, singletons were retained in the dataset as possible representatives of low-expression transcript fragments. Differential expression analysis was set with a minimum length fraction = 0.6 and a minimum similarity fraction (long reads) = 0.5. 

Contig sequences obtained from each tissue were blasted to CGE regions to enrich the number of transcripts evaluated by RNA-Seq analysis (Figure 1). In addition, the sequences were extracted from the Atlantic salmon genome near to the threshold areas in a window of 10 kb for each transcriptome. The expression value was set as transcripts per million (TPM). The distance metric was calculated with the Manhattan method, with a mean expression level in 5–6 rounds of k-means, clustering subtracted. Finally, Generalized Linear Model (GLM) available in the CLC software was used for statistical analyses and to compare gene expression levels in terms of the log_2_ fold change (*p* = 0.005; FDR-corrected). In addition, k-means clustering was performed for transposable element (TE) expression values. The metric distance was calculated with the Manhattan method, where the mean expression level in 5–6 rounds of k-means clustering was subtracted.

### 2.5. Sequence Annotation and GO Enrichment Analysis

Differentially expressed contigs were annotated through BlastX analysis using a custom protein database constructed from GeneBank and UniProtKB/Swiss-Prot. The cutoff E-value was set at 1E-10. Transcripts were subjected to gene ontology (GO) analysis using the Blast2GO plugins included in the CLC Genomics Workbench v12 software (CLC Bio, Qiagen, Germantown, MA, USA). The results were plotted using the cluster Profiler R package [32]. GO enrichment analysis was conducted to identify the most represented biological processes among protein-coding genes located proximally to the identified lncRNAs. Enrichment of biological processes was identified using Fisher’s exact test tool of Blast2GO among the different experimental groups against the control group (FW samples). 

### 2.6. miRNA Annotation and Expression Analysis in Response to Salinity Changes

Low-quality reads from illumina sequencing data, reads with a quality score of less than 0.05 on the Phred scale, with a short length, or with three or more ambiguous nucleotides were removed using CLC Genomics Workbench software (Version 21, CLC Bio, Aarhus, Denmark). Furthermore, any cleaned sequences matching metazoan mRNA, rRNA, tRNA, snRNA, snoRNA, repeat sequences, or other ncRNAs were deposited in the NCBI databases (http://www.ncbi.nlm.nih.gov/ (accessed on 27 July 2021)), RFam (http://rfam.janelia.org/ (accessed on 27 July 2021)), or Repbase (http://www.girinst.org/repbase/ (accessed on 27 July 2021)) were discarded. Then, the remaining transcripts were counted to generate a unique small RNA list. These sequences were annotated against pre-miRNA and mature miRNA (5′ and 3′) sequences listed for *Salmo salar* available in the miRbase (release 22) [16,33]. miRNA expression analysis followed a similar protocol, as previously described by our group [34]. 

### 2.7. miRNA Target Prediction and Expression Correlation 

The computational target prediction algorithm used was RNA22 [35]. The datasets used were the differently modulated transcripts from both groups, GSC fish and SS fish. The RNA22 parameters were set at free energy < −15 kcal/mol and a score > 50. All target genes were annotated by GO analysis, following the protocol described above.

In addition, smoltification-related genes and differently expressed miRNAs were selected for expression-correlation analysis. Pearson’s correlation among TPM values from both datasets, including all samples exposed and not exposed to each salinity condition, were calculated in R software [36]. Plots for correlation analyses were constructed using the Corrplot package [37], considering a correlations with *p*-value < 0.01 significant. 

### 2.8. RT-qPCR Validation Analysis

Transcription expression profiles of genes associated with the smoltification process and ncRNAs were conducted by RT-qPCR. Briefly, 200 ng/μL of total RNA from three individuals per fish group obtained was used for cDNA synthesis using the RevertAid™ H Minus First Strand cDNA Synthesis Kit (Thermo Fisher Scientific™, USA), following the manufacturer’s instructions. Four genes and their putative miRNAs were validated (Appendix A). The comparative ΔΔCt relative expression analysis method was used. Selection of the housekeeping gene for the experiment was based on evaluation of the stability of elongation factor-ɑ (EF-ɑ), β-tubulin, and 18S genes by Normfinder. Here, EF-ɑ was selected for gene normalization. Each RT-qPCR reaction was carried out in a final volume of 10 μL using the commercial PowerUp SYBR Green Master Mix kit (Applied Biosystems^®^, Waltham, MA, USA). RT-qPCR reactions were performed on the StepOnePlus™ (Applied Biosystems^®^, Life Technologies™, Carlsbad, CA, USA) using the following conditions: 95 °C for 10 min, 40 cycles at 95 °C for 15 s and 60 °C for 30 s, ending with 30 s at 72 °C. Statistical analyses were conducted through ANOVA-1 test and Student’s *t*-test in the GraphPad Prims 8.4.7 package.

miRNA transcription-level validation was achieved by synthetizing cDNA from the same RNA samples using the miSCript II RT kit (Qiagen, Germany), with an incubation reaction at 37 °C for 60 min and 5 min at 95 °C. Specific primers were designed for bantam miRNA and were used for amplification by qPCR using the miScript SYBR Green PCR kit (Qiagen, Germany) in a QuanStudio 3 System (Life Technologies, USA). Thermal cycling conditions consisted of an initial denaturation and enzyme activation at 95 °C for 15 min, followed by 40 cycles of 94 °C for 15 s (denaturation), 55 °C for 30 s (annealing), and 70 °C for 45 s (extension). Ssa-mir-455e5p was used as an endogenous control for this reaction [38]. Gene and miRNA expression were quantified using the ΔΔCT comparative method, was previously described (Pfaffl, 2001).

## 3. Results

### 3.1. Atlantic Salmon Performance during the Experimental Trial

Immune histochemistry analysis performed in Atlantic salmon exposed to GSC and SS conditions showed chloride cell migration, indicating adequate salmon conditions for SW transfer (Appendix A). Furthermore, this condition was confirmed by RT-qPCR analysis of ATPase-α and ATPase-β subunits (Appendix A). No mortality was recorded in experimental groups.

### 3.2. Gill-Tissue-Transcription Modulation during Smoltification Process

Whole-genome modulation of Atlantic salmon tissues was evaluated in two groups: Atlantic salmon exposed to gradual salinity change (GSC) and salinity shock (SS). Atlantic salmon gill-tissue whole-genome expression showed low variation in MRNAs between the GSC and SS fish groups (Figure 2A). Moreover, miRNAs had an opposite expression pattern compared with mRNAs, with high threshold values in chromosome areas where mRNAs showed a downregulation. Chromosome expression variation between experimental groups was calculated by CGE index. Despite the low differences in mRNA threshold values between the GCS and SS groups, a group of chromosomes including chr1, chr2, chr4, chr9, chr12, chr13, chr14, chr17, chr18, chr22, and chr25 presented high CGE index values (Figure 2B). In addition, these chromosomes showed high threshold values for miRNAs, with a high CGE index. Furthermore, the synteny analysis of the selected chromosomes exhibits a section with high homology among the highlighting areas (Figure 2B). Notably, chr12 and chr22 showed a large synteny block and high mRNA modulation (red ribbons), and similar patterns were found in chr12 with chr2 in green ribbons. This also draws attention to chr4 and chr13, with high miRNA modulation linking a synteny block.

Putative differentially expressed chromosome regions were annotated by Blast analysis using a *Salmo salar* protein database. Notably, numerous transposase and transposable elements, *Tcb1*, *HSP70*, and *MCHII* genes were annotated in chromosomes showing high differential expression among experimental fish groups (Appendix A). Moreover, RNA-Seq analyses of these selected chromosomes evidenced an interesting expression pattern, where gill samples of FW and fish from the SS group at 32 PSU were grouped in the same cluster, separated from the GSC 32PSU group (Figure 2C). Furthermore, differential expression analysis of transposable elements (TEs) showed that the median of the TEs expressed in fish exposed to GSC was downmodulated (Appendix A).

In addition, contigs annotated as TEs clustered in the Atlantic salmon gill transcriptome. Fish transfer from FW to SW gradually and by salinity shock exhibited different expression (Appendix A). Notably, clusters 1 and 4 were downmodulated after SW transfer at 32 PSU, GSC group. In contrast, cluster 2 showed contigs annotated as upmodulated TEs (Appendix A). Furthermore, gill samples of Atlantic salmon from FW to 32 PSU by salinity sock showed four clusters, where cluster 1 and 3 showed different expression profiles, up- and down-modulated, respectively. Interestingly, TEs RT-qPCR validation showed a similar expression pattern to RNA-seq analysis in cluster 2 and cluster 3 for GSC and SS, respectively (Appendix A).

### 3.3. Intestine Tissue-Transcription Modulation during Smoltification Process

A high regulation of the expression of chr2 to chr9 regions was observed from the whole-genome expression analysis. Moreover, miRNA transcriptional variation was observed in a reduced number of chromosomes, including chr4, chr13, chr15, and chr25 (Figure 3A), suggesting a low regulatory role of miRNAs in the intestine during the smoltification process. Notably, chromosomes with a high CGE index between mRNAs expressed in the GSC and SS fish group showed a high miRNA CGE index (Figure 3B), suggesting local miRNAs regulation, similar to the findings observed in gills. In addition, the synteny analysis among highlighting chromosomes showed an area with a high CGE index and mRNA homology between chr1-chr9 and chr12-chr22. Contrary to was observed in gills, heatmap representations of intestine transcripts group in the same cluster fish at 32 PSU (Figure 3C). In addition, intestine chromosomes with a high CGE index exhibited a great abundance of regulatory elements, such as transposase, transposable elements, and retro-transposable elements (Appendix A).

### 3.4. Head-Kidney Transcriptome Modulation during Smoltification Process

From chromosome analysis using read sequences obtained for head kidney tissue from Atlantic salmon exposed to GSC and SS, high modulated areas were observed in all Atlantic salmon genomes (Figure 4A). Additionally, head-kidney miRNA analysis of whole Atlantic salmon genome suggested an important regulatory function during the smoltification process in this tissue. Cromosomes chr2, chr3, chr4, chr9, chr13, chr15, chr17, chr20, chr24, and chr25 had high CGE indexes for mRNA and miRNA between head kidney tissue of fish exposed to GSC and SS (Figure 4B). Interestingly, while the synteny analysis of gill and intestine tissue showed homology among mRNAs areas, a synteny block in head kidney tissue was observed in regions with a high miRNA CGE index, such as chr13-chr15, chr20-chr24, chr3-chr6, and chr13-chr14 (Figure 4B). Interestingly, among the genes annotated in these chromosomes, *hemoglobin*, *HSP70*, *TLR8*, and a large number of transposase and transposable elements were found (Appendix A). Notably, TEs expressed in head kidney were upregulated in both experimental conditions, GSC and SS (Appendix A).

### 3.5. Differential Expression Analysis of Atlantic Salmon during Seawater Transfer

Nucleotide sequences near 10kb of each coverage threshold area were extracted from the Atlantic salmon genome. Later, sequences were blasted to the contigs obtained from the de novo assembly performed for each tissue. These sequences were used as a reference for RNA-Seq analysis using the filtered data of each tissue. Gill tissue of fish exposed to GSC and SS at 32 PSU showed similar expression patterns. On the other hand, samples obtained from the control group (FW) and fish exposed to 10 and 20 PSU were grouped in the same cluster (Appendix A). From the differential expression analysis between gills of fish exposed to GSC and SS compared with the control group (FW), a larger number of transcripts differently modulated in response to the SS, with 2528 transcripts, compared to 646 transcripts differently expressed in the GSC group, suggesting a signification effect of SS in mRNA expression modulation (Figure 5A, Appendix A). From GO analysis, the differentially expressed transcripts were annotated as biological processes, such as response to a steroid hormone, positive regulation of gene expression, muscle cell migration, localization of cell, insulin secretion, defense response, eye morphogenesis, and cell motility, with a large number of transcripts in the GSC group (Figure 5A).

Similar expression patterns were found between intestine fish samples from GSC and SS under the 32 PSU condition. In the control group (FW), fish exposed to 10 and 20 PSU were clustered in the same group (Appendix A). A total of 3528 transcripts were highly modulated in the intestine of the GSC group, while in SS fish group, 1483 DE transcripts in intestine tissue were obtained (Figure 5B, Appendix A). GO enrichment analysis of differently modulated transcripts in intestine tissue showed BP processes associated with response to hormones, organic substance metabolic processes, metabolic processes, ion transmembrane transport, cellular processes, and some processes associate with immune response, with a greater number of transcripts in the GSC group (Figure 5B).

Transcriptome expression analyses showed a similar expression profile between head-kidney samples of fish at 32 PSU from GSC and SS groups. Notably, the head-kidney samples obtained from the GSC fish group at 10 PSU showed high expression levels of transcripts downregulated under the other evaluated conditions (Appendix A). Interestingly, differential expression analysis showed a large number of transcripts modulated in the SS group, with 2480, compared with to 681 transcripts modulated in response to the GSS condition (Figure 5C, Appendix A). GO analysis showed that the most relevant modulated BP were in response to oxidative stress, response to growth factor, positive regulation of transcription, and positive regulation of response to extracellular stimulus, among others (Figure 5C).

### 3.6. miRNA Regulation in Atlantic Salmon during Smoltification

The small RNAs obtained from Illumina sequencing for each tissue were annotated using the *Salmo salar* miRNA database published in miRBase release 22.1 [16,33]. A total of 478 miRNAs were annotated, like those reported in the miRBase for Atlantic salmon. In gills, two clusters of transcriptional profiles were identified, the first one grouping the control group (FW) with fish exposed to 10 PSU, and a second cluster grouping samples obtained from fish exposed to GSC at 20, 32 PSU and the SS group at 32 PSU (Figure 6A). Unlike gills, the transcriptional patterns of miRNAs of SS and GSC were grouped by salinity (32 PSU) in the same cluster. Furthermore, in intestine and head kidney tissue, fish exposed to GSC (32 PSU) evidenced an upregulation compared to fish from the SS group. In contrast, GSC fish at 20 and 10 PSU were fish sampled in freshwater (Figure 6B,C).

Notably, clustering analysis showed different miRNAs with expression profile changes related to the FW condition between GSC and SS groups. Interestingly, miRNAs with expression changes between FW and GSC showed an upregulation in gill samples after transferring to 32 PSU (Appendix A). Furthermore, these miRNAs expressed in response to GSC, ssa-miR-143, ssa-miR-21b, and ssa-miR-10d, were validated by RT-qPCR. Expression evaluation showed an up-modulation during salinity changes from FW to 32 PSU, similar to the in silico analysis (Appendix A). Furthermore, four cluster exhibited significant expression changes in fish groups exposed to SS (Appendix A). In addition, RT-qPCR analysis in gill samples obtained at each sample point (Appendix A) showed similar expression patterns of ssa-miR-181 and ssa-miR-30d from cluster 1, and ssa-miR-10b from cluster 2 (Appendix A).

Differential expression analysis was performed between GSC and SS salmon groups using sampled tissues in FW as a control. Differences in the number and class of miRNAs among tissues were found. For instance, gill samples showed a large number of miRNAs differently expressed in response to SS, including SSA-miR-122-5p, ssa-miR-122-3p, and ssa-miR-122-2-3p upregulated in GSC group. Moreover, according to prediction target analysis, SSA-miR-122-5p has as target gene the *sodium/potassium-transporting ATPase subunit beta-3-like* and *thyroid hormone receptor interactor 11*, with delta G values of −17.3 and −16.2, respectively, While SSA-miR-365-5p is the most downregulated miRNA in this tissue (Figure 7).

A larger number of miRNAs in the intestine were differently expressed in samples obtained from fish exposed to GSC. However, no significant differences in the expression change values were observed (Figure 7). However, upregulated SSA-miR-155-3p miRNA showed a putative binding site to *myosin-11-like*, *probable cation-transporting ATPase*, and *collagen Type XI Alpha2* in the GSC group, with delta G values of −14.5, −13.5, and −14.7, respectively (Appendix A). Moreover, ssa-miR-20a-1-3p was upregulated, and among its target genes, the *membrane heat shock 70 kDa protein* and *toll-like receptor 13* were identified, with delta G values of −12.54 and −18.2, respectively (Appendix A).

The head kidney showed a similar number of differently expressed miRNAs between CGS and SS fish groups. Nevertheless, the miRNAs differently modulated between groups were different. For instance, SSA-miR-499a-5p, ssa-miR-192a-3p, ssa-miR-192b-3p, and ssa-miR-200b-5p were overexpressed in the GSC group. Target gene analysis determined that cathelicidin antimicrobial peptide has a putative binding site to ssa-miR-192a-3p, with a delta G value of −15.2, while ssa-miR-122-5p and ssa-miR-122-2-3p were upregulated in response to SS, evidencing a putative binding site to *immunoglobulin tau heavy chain* (delta G −15.2) (Appendix A).

Interestingly, some miRNAs are expressed in two tissues, suggesting a role in salinity changes. For instance, ssa-miR-499a-5p miRNAs was overexpressed in gills and head kidney tissues in response to GSC. Another example is ssa-miR-196b-3p, which is overexpressed in the intestine and head kidney. Regarding the response to SS, the miRNAs ssa-miR-27d-5p is upmodulate in the gills and intestine. Others interesting miRNAs are ssa-miR-144-3p and ssa-miR-301a-5p, downregulated in gills and head kidney from fish exposed to SS (Figure 7). Furthermore, target analysis of commonly expressed miRNAs evidenced a putative role in the modulation of collagen genes in the case of ssa-miR-499a-5p and ssa-miR-27d-5p. Additionally, it was a gene associated with cell differentiation was observed as *G-protein-signaling modulator 2-like*, with a target site to ssa-miR-144-3b. Finally, among the SSA-miR-301a-5p target genes was identified *clathrin heavy chain 1-like*. (Appendix A).

### 3.7. GO Enrichment of miRNA Target Genes

The differentially expressed mRNAs in each tissue of fish groups exposed to GSC and SS were evaluated as target genes of differently modulated miRNAs. Notably, GO enrichment analysis of putative mRNAs suggests that gradual salinity changes trigger a greater number of regulatory responses than the salinity shock in the evaluated tissues. This is reflected in BP number, which seems to be modulated by differential transcribed miRNAs in the GSC fish group (Table 1). Response to stimulus, cell communication, response to stress, and immune system process were found among the Biological Process (BPs) putatively modulated by miRNAs in the three tissues (Table 1). The fish group exposed to SS exhibits a reduced number of BPs putatively modulated by differentially expressed miRNAs (Table 1). Furthermore, among the tissues, different target processes were identified. For instance, among the most representative processes in gills were pattern-recognition receptor-signaling pathways and response to ATP. On the other hand, regulation of mononuclear cell proliferation and antigens processing a presentation by MHC class II were observed in the intestine. While, the BPs found in head kidney were glycosylation and transposition (Table 1).

In particular, the evaluation of expression changes of miRNAs and their putative targets by tissue suggested a putative regulatory role of ssa-miR-205-5p in the regulation serine/threonine kinase or the regulation of the HSP70 gene by ssa-moR-19c-3-5p in gills exposed to GSC (Table 2, Appendix A). On the other hand, in the case of SS fish gills, a putative regulatory role of ssa-miR-19d-5p and ssa-miR-222b-5p with ATPase inhibitor can be mentioned and transposase, respectively (Table 2). Moreover, a putative regulation of *myosin* and *annexin A2-like* genes in the intestine tissue of GSC fish was observed by SSA-miR-92a-5p and ssa-miR-15a-3p, respectively. In addition, in SS fish group was observed a negative correlation expression between *free fatty acid receptor* and *low-density lipoprotein receptor* genes, with miRNAs 30a-3p and 125b-5p, respectively (Table 2). In the case of head kidney tissue of GSC fish, ssa-miR-128-1-5p showed a negative correlation expression with the *SPATA5*gene, and *laminin* gene exhibited a binding site to ssa-miR-194a-3p in fish exposed to GSC (Table 2).

Finally, expression-correlation analysis was conducted among the differently expressed miRNAs and smoltification-related genes and TEs to evidence the investigated role observed during smoltification. Expression-correlation analysis exhibited a negative correlation value between ssa-miRNA-204-5p with the *TE Tcb1* and the *sodium/potassium-transporting ATPase subunit alpha and beta.* Additionally, TEs showed negative correlation values with miRNAs ssa-miRNA-19c-3-5p, ssa-miRNA-138-5p, and ssa-miRNA-10a-2-3p (Figure 8). Interestingly, positive correlation values were found in some isoforms of *sodium/potassium-transporting ATPase subunit alpha and beta* and the miRNAs belonging to the families ssa-miRNA-206, ssa-miRNA-214, and ssa-miRNA-219. This result could be associated with specific regulatory roles between miRNAs and transcript isoforms. 

## 4. Discussion

This study proposes a novel approach to evaluate transcriptome data, where time-series of gene-expression profiling from different tissues and experimental conditions are evaluated. Here, the whole-genome expression-profiling of mRNAs and miRNAs of Atlantic salmon during the smoltification process were explored. Furthermore, the methodology to determine the chromosome gene expression (CGE index) was described. CGE index indicates the coverage differences among two conditions along the Atlantic salmon genome, reflecting differences in the number of mapped reads in specific genomic regions. The study included three different Atlantic salmon tissues with osmoregulation function —gills, intestine, and head kidney. Fish were exposed to gradual salinity changes and salinity shock.

Interestingly, synteny analysis shows high homology among chromosomes with a high CGE index. Furthermore, these chromosomes showed a large number of mobile genetic elements as transposable elements (TE) from Tc1 family and transposase genes. The transposable elements are highly represented in the Atlantic salmon genome. For instance, only the transposable elements of the Tc1-mariner family class represent the 12.89% of the genome [1]. Moreover, this group of transposable elements was associated with signal transduction, regulation of transcription, and defense response [39]. Additionally, it has been reported that transposon expression in rainbow trout is triggered by an external stimulus, such as stress, toxicity, or bacterial antigens [40]. In addition, the growth hormone gene highly associated with the smoltification process, a transposon insertion in a specific isoform of growth hormone gene (gh2) promotor of Atlantic and chinook salmon, has been described. Furthermore, the authors identified a second Tc1 transposon only in the gh2 gene of Atlantic salmon, associated with speciation events [41]. Notably, a previous study performed for our group reported a high presence of TE located near lncRNAs with a putative role in Atlantic salmon seawater adaptation [9]. We suggest a strong regulatory event associated with TEs during Atlantic salmon smoltification. Moreover, this study reports upregulation of TEs in head kidney tissue of Atlantic salmon exposed to GSC and SS. In contrast, gill tissue showed upregulation of TEs in fish exposed to SS, suggesting a putative role of TEs in salmon seawater adaptation. However, additional functional studies are needed to demonstrate the role of TEs during the smoltification process.

Differential expression patterns during parr-smolts transition have been reported by cDNA microarray analysis, highlighting up-modulation of genes associated with growth, metabolism, oxygen transport, and osmoregulation [7]. Additionally, upregulation of genes related to transcription, oxygen transport, electron transport, and protein biosynthesis has been reported [8]. Furthermore, an RNA-Seq study of Atlantic salmon gills showed expression variation during gradual salinity changes, showing higher modulation of processes associated with stress response, cell division and proliferation, tissue development, and collagen catabolic process [9]. Additionally, significant enrichment of genes related to immune response, response to stress, and growth have been described in Atlantic salmon smolt head kidney tissue following seawater transfer [18]. In addition, the authors reported an immune response downregulation after a week in seawater [42]. Moreover, a downregulation of the immune system has been suggested by Shwe, Østbye, Krasnov, Ramberg, and Andreassen [18], associated with the pathogen susceptibility reported for Atlantic salmon during the first period in seawater [42].

Interestingly, a remarkable difference in the number of transcripts differently expressed (DE) among fish exposed to GSC and SS was observed in the present study. It is possible to suggest that salinity shock triggers a higher transcription response than GSC, where the number of DE transcripts was low. GO enrichment analysis of evaluated tissues exhibit BPs associated with protein metabolic process, localization and cell motility, biosynthesis, metabolism, immune response, response to oxidative stress, and response to growth factor. Interestingly, eye morphogenesis was also identified among GO terms identified in gills tissue. In vertebrates, the eyes have a relevant osmoregulatory role [43]. In Coilia nasus, expression variation of eye transcriptomes was observed between hyperosmotic and hypoosmotic conditions. Among the differentially regulated genes were annotated genes associated with immune response, metabolism, and transport [43].

Previously, our group characterized long, non-coding RNAs of Atlantic salmon gills during the smoltification process and reported on the relevance of non-coding RNAs in regulation of this biological process [9]. Interestingly, the study observed that highly regulated lncRNAs were located near genes associated with the process as growth, cell death, catalytic activity, and apoptotic process [9]. The role of miRNAs during smoltification and the early seawater phase was described by Shwe, Østbye, Krasnov, Ramberg, and Andreassen [18] in head kidney of Atlantic salmon. The authors reported 71 DE miRNAs during their evaluation time. Among the relevant miRNAs identified by the authors, the miRNA from family mir-192 has been associated with hypoxia [44]. In the present study, the mir-192 family was overexpressed in response to GSC in head kidney and down modulated in gills of the SS fish group. Another miRNA family associated with hypoxia [44], and downregulated in the intestine in our study was mir-181. Notably, in the present study, SSA-miR-499a-5p was up-regulated in gills and head kidney tissues of Atlantic salmon exposed to GSC. Moreover, from the target analysis, it is possible to suggest that miRNA has a role in fish growth, with gene collagen as its target. In contrast, miRNA was downregulated in head kidney of the SS fish group. Similar results were reported by Shwe, Østbye, Krasnov, Ramberg, and Andreassen [18], who observed a decrease in ssa-miR-499a-5p expression in Atlantic salmon head kidney during transfer from FW to SW.

Finally, GO enrichment analysis of putative target genes of differently modulated miRNAs from the GSC group showed a large number of transcripts associated with cellular process, response to stimulus, cell communication, response to stress, and immune system process in gills, intestine, and head kidney tissues. A similar process was reported to be associated with target genes of miRNAs differently expressed in Atlantic salmon head kidney during seawater adaptation [18]. In contrast, in this study, a small number of target genes was annotated in response to salinity shock. For instance, biological process as a response to A4TP, antigens processing a presentation by MHC class, and glycosylation were identified in this study. These results suggest bounded Atlantic salmon miRNA modulation in response to salinity shock, compared with gradual salinity change, where Atlantic salmon miRNAs display regulation of a large number of biological processes. Functional analysis will be conducted to demonstrate the regulatory role of the highlight miRNAs identified in this study.

## 5. Conclusions

The CGE index proposed in this study explains differences in expression profiling among fish exposed to gradual salinity change and salinity shock during seawater transfer. Furthermore, a great abundance of transposable elements was identified among the chromosomes, with significant expression differences between groups. This is the first study showing the harmony among mRNA and miRNA transcription profiles at the genome level of Atlantic salmon during the smoltification process. The proposed transcriptome analysis revealed significant differences among tissues, time, and experimental conditions. In addition, the relevance of miRNA function in the regulation different biological process is suggested, such as growth, stress response, catabolism, and immune response.

## Figures and Tables

**Figure 1 biology-11-00001-f001:**
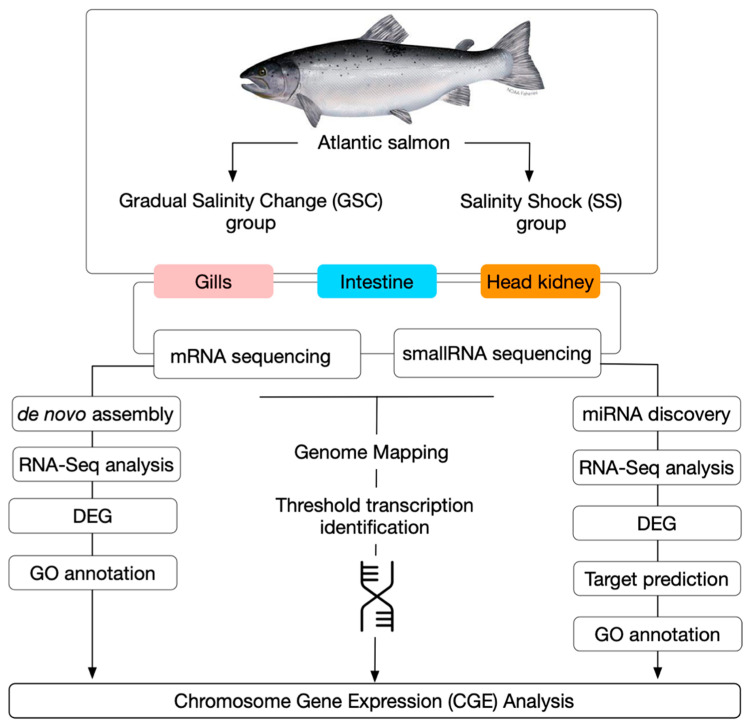
Whole-genome expression approach for transcriptome analysis during smoltification in Atlantic salmon.

**Figure 2 biology-11-00001-f002:**
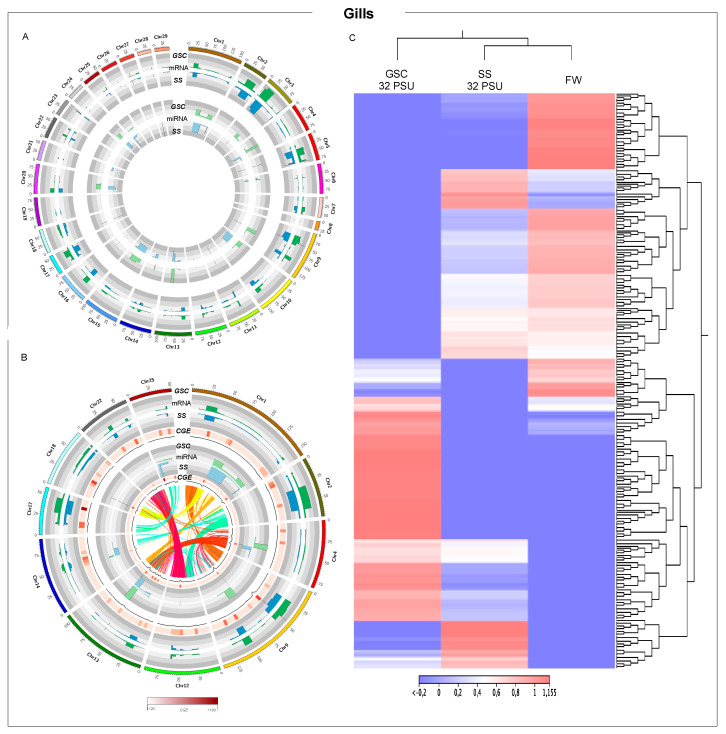
Whole-genome transcription of Atlantic salmon gills during smoltification process. (**A**) Threshold analysis of gills for GSC and SS fish groups. (**B**) Chromosome regions with high CGE index variation between GSC and SS fish groups. Heatmap in red shows the expression variation between both groups, CGE index. Black line graph indicates genome coverage of threshold areas. In the Circos plot, the ribbons represent the homoeologous regions in salmon genome. (**C**) RNA-Seq analysis of chromosome regions with high CGE index between experimental groups.

**Figure 3 biology-11-00001-f003:**
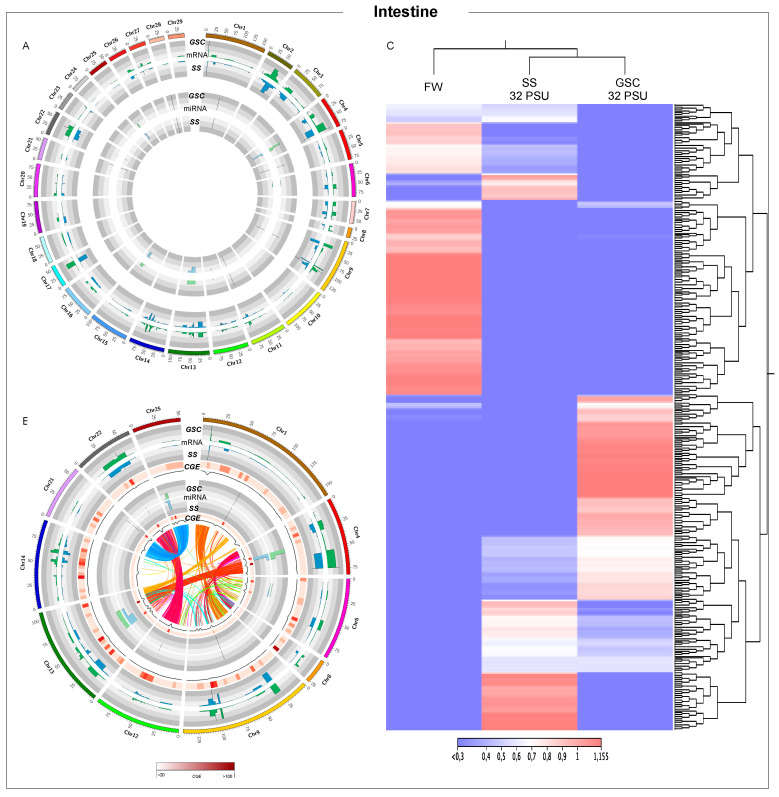
Whole-genome transcription in Atlantic salmon intestine during smoltification process. (**A**) Threshold analysis of gills for GSC and SS fish groups. (**B**) Chromosome regions with high CGE index variation between GSC and SS fish groups. Heatmap in red shows the expression variation between both groups, CGE index. Black line graph indicates genome coverage of threshold areas. In the Circos plot, the ribbons represent the homoeologous regions in salmon genome. (**C**) RNA-Seq analysis of chromosomes regions with high CGE index between experimental groups.

**Figure 4 biology-11-00001-f004:**
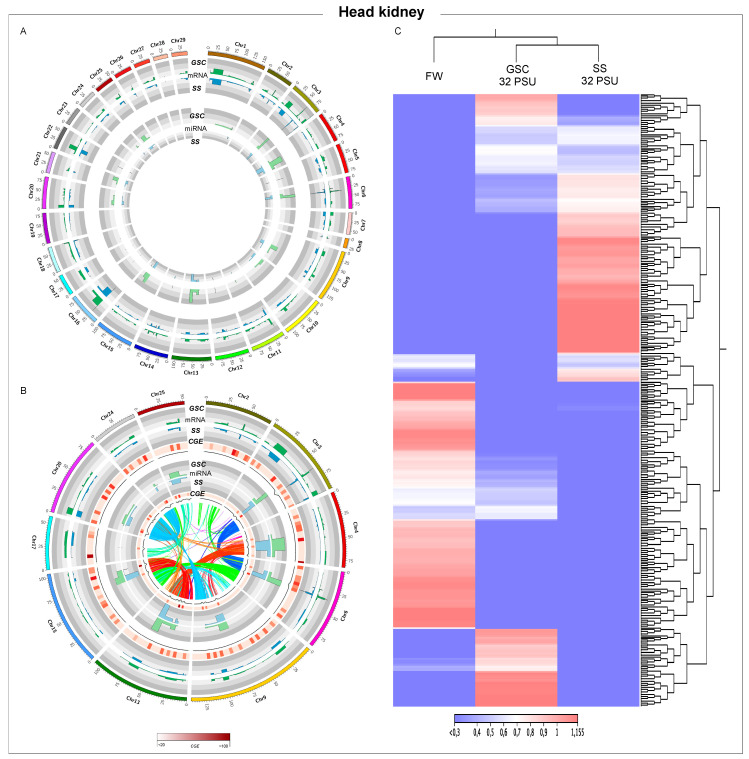
Whole-genome transcription in Atlantic salmon head kidney during smoltification process. (**A**) Threshold analysis of gills for GSC and SS fish groups. (**B**) Chromosome regions with high CGE index variation between GSC and SS fish groups. Heatmap in red shows the expression variation between both groups, CGE index. Black line graph indicates genome coverage of threshold areas. In the Circos plot, the ribbons represent the homoeologous regions in salmon genome. (**C**) RNA-Seq analysis of chromosome regions with high CGE index between experimental groups.

**Figure 5 biology-11-00001-f005:**
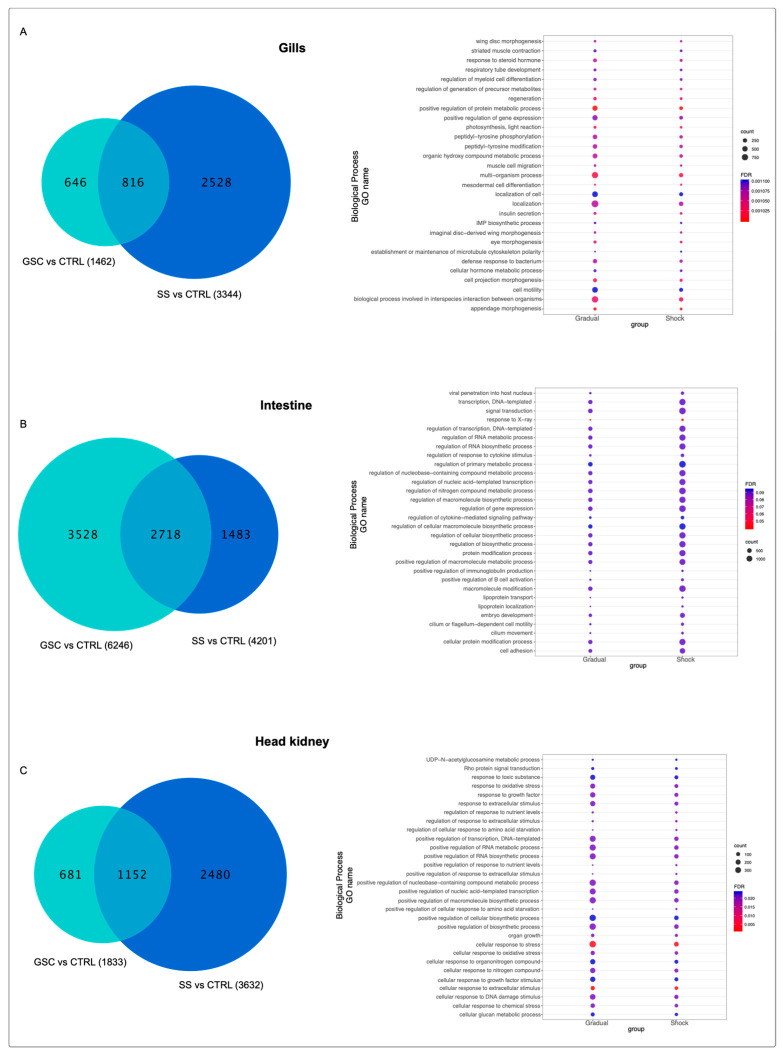
Differential expression analysis and GO enrichment of Atlantic salmon exposed to GSC and SS. (**A**) DEGs and GO enrichment of gills tissue. (**B**) DEGs and GO enrichment of intestine tissue. (**C**) DEGs and GO enrichment of head kidney tissue.

**Figure 6 biology-11-00001-f006:**
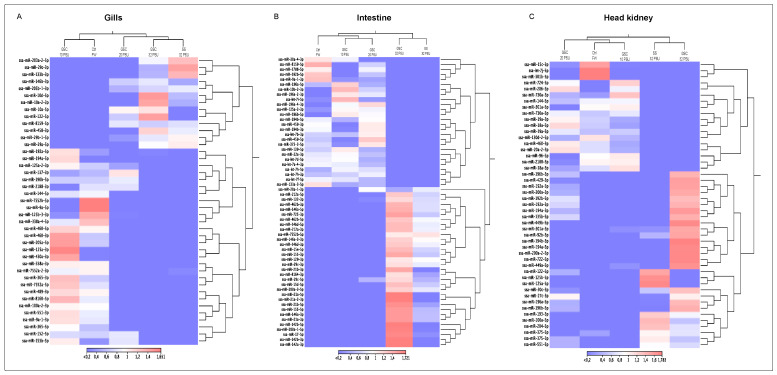
miRNA expression profile during gradual salinity changes and salinity shock in gills, intestine, and head kidney of Atlantic salmon.

**Figure 7 biology-11-00001-f007:**
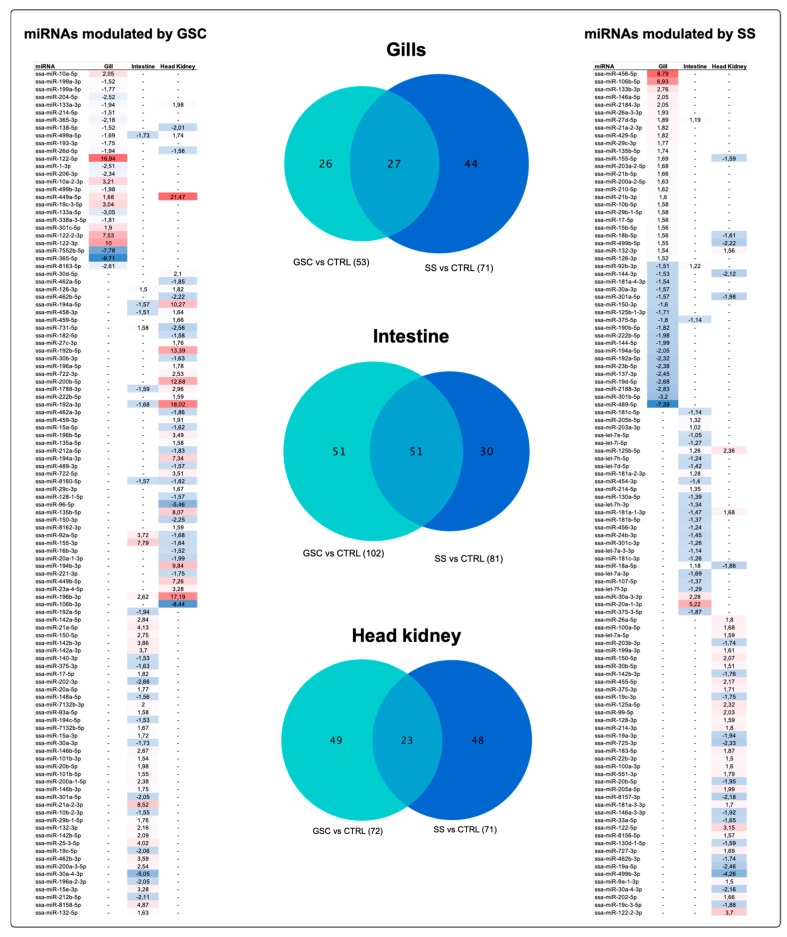
miRNA differential expression analysis of Atlantic salmon tissues under both conditions, GSC and SS. Tables show the fold-change values of miRNA for each tissue; red: upregulated, blue: downregulated.

**Figure 8 biology-11-00001-f008:**
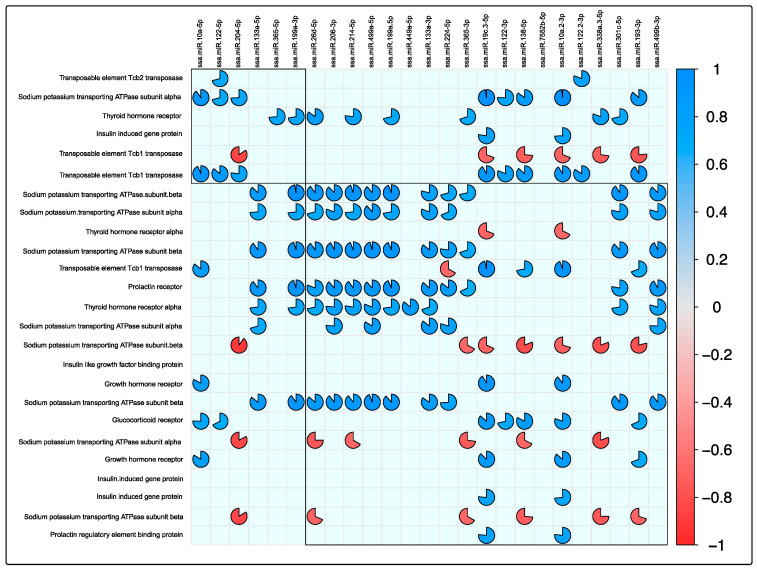
Correlation analysis of expression pairs among smoltification-related genes, TEs, and differential expressed miRNAs. TPM values. Corrplot analyses were conducted on differentially expressed smoltification-related genes, TEs, and differential expressed miRNAs (fold change > |4| and *p*-value < 0.01) in the combination of all the data, including exposure to the three tissue and GSC and SS conditions. Only Pearson’s correlation values that were significant (*p*-value > 0.01) are shown in the plot. Red pies correspond to significant negative correlations, and blue pies correspond to significant positive correlations. The completeness of the pies corresponds to the correlation level, where pies closer to circular shape correspond to values more proximal to |1| in Pearson’s calculation.

**Table 1 biology-11-00001-t001:** GO enrichment analysis of differently modulated putative miRNA target genes in Atlantic salmon exposed to gradual salinity changes (GSC) and salinity shock (SS).

GSC Atlantic Salmon		SS Atlantic Salmon	
Gill Tissue Biological Process	N° GO Term	Gill Tissue Biological Process	N° GO Term
Cellular process	269	Pattern recognitio receptor signaling pathway	126
Metabolic prosess	259	Regulation of Pattern recognitio receptor signaling pathway	121
Response to stimilus	243	Truptophanil-tRNA aminoacylation	59
Macromolecule metabolic process	241	Protein ADP-ribosylation	53
Biological regulation	239	Regulation od protein ADP-ribosylation	51
Regulation of biological process	234	Spindle assembly	43
Cellular biosynthetic process	233	Mitotic spindle assembly	32
Regulation of cellular process	232	Response to ATP	30
Organic cyclic compound metabolic process	229	Regulation of meiotic cell cycle	28
Celullar response to stimulus	226	Microtube nucleation	23
Gene expression	221	Carbon utilization	22
Regulation of metabolic process	218	Regulation of mitotic spindle assembly	20
Localization	215		
Cell communication	214		
Transport	212		
Organic ciclyc compound biosynthetic process	210		
Regulation of cellular metabolic process	210		
Cellular component organization	209		
Cellular component organization or biogenesis	209		
Portein metabolic process	209		
Cellular macomolecule biosynthetic process	208		
Response to stress	203		
Immune system process	202		
Small molecules metabolic process	202		
Regulation of gene expression	202		
**Intestine Tissue Biological Process**	**N° GO Term**	**Intestine Tissue Biological Process**	**N° GO Term**
Cellular process	269	Regulation of mononuclear cell proliferation	8
Metabolic prosess	259	Regulation of lymphocyte proliferation	8
Organic substance metabolic process	256	Antigen processing and presentation of peptide	8
Cellular metabolic process	255	Hydrogen peroxide catabolic process	7
Prymary metabolic process	253	Antigen processing and presentation of peptide antigen	7
Response to stimilus	243	MCH class II	6
Macromolecule metabolic process	241	Fertilization	5
Biological regulation	239	Cell-cell recognition	4
Organic substance biosynthetic process	237	Sperm-egg recognition	3
Biosynthetic process	237	Lung epithelium development	3
Regulation of biological process	234	Hemolysis in other organims	3
Cellular macomolecule metabolic process	233	Microtubule polymerization	3
Regulation of cellular process	233	Regulation of platelet activation	3
Cellular response to stimulus	232	Forebrain neuron differentation	3
Gene expression	226	Forebrain generation of neurons	3
Nucleobase-containing compund metabolic process	221		
Regulation of metabolic process	221		
Cellular nitrogen compound bisynthetic process	218		
Localization	215		
Cell communication	214		
Transport	212		
Response to stress	203		
Immune system process	202		
Regulation of gene expression	202		
**Head Kidney Tissue Biological Process**	**N° GO Term**	**Head Kidney Tissue Biological Process**	**N° GO Term**
Cellular process	169	DNA integration	82
Metabolic prosess	165	Transposition	55
Cellular metabolic process	163	Transposition, DNA-mediated	55
Organic substance metabolic process	163	Macromolecule glycosylation	42
Primary metabolic process	158	Protein glycosilation	42
Nitrogen compund metabolic process	154	Pryrimidine nucleotide-sugar transmembrane transport	19
Cellular biosynthetic process	153	Nucelotide-sugar transmembrane transport	19
Biosyntetic process	153	ARF protein signal transduction	10
Organic substance biosynthetic process	152	Regulation of ARF protein signal transduction	10
Macromolecule metabolic process	151		
Biological regulation	149		
Organonitrogen compound metabolic process	149		
Response to stimilus	147		
Cellular macomolecule metabolic process	147		
Regulation of biological process	146		
Localization	144		
Cellular nitrogen compound metabolic process	143		
Regulation of cellular process	143		
Organic cyclic compound metabolic process	143		
Transport	142		
Establishment of localization	142		
Immune system process	124		
Immune response	119		
Defense response	108		
Response to biotic stimilus	103		

**Table 2 biology-11-00001-t002:** Expression modulation of miRNAs and their putative target genes in Atlantic salmon exposed to GSC and SS.

Tissue	miRNA	Fold Change GSC	Fold Change SS	*De nodo* Assembly Contig	Fold Change GSC	Fold Change SS	Delta G	Description
Gills	ssa-miR-204-5p	−2, 52	0	contig_33389	2, 14	0	−12.00	PREDICTED: serine/threonine-protein kinase WNK2-like isoform X2 [Salmo salar]
	ssa-miR-19c-3-5p	3, 04	0	contig_41523	−2, 29	0	−12.00	PREDICTED: heat shock 70 kDa protein 12B-like [Salmo salar]
	ssa-miR-214-5p	−1, 51	0	contig_50634	2, 59	0	−12.00	PREDICTED: creatine kinase S-type, mitochondrial isoform X1 [Salmo salar]
	ssa-miR-214-5p	−1, 51	0	contig_29197	2, 47	0	−12.20	PREDICTED: MAPK/MAK/MRK overlapping kinase-like isoform X1 [Salmo salar]
	ssa-miR-199a-3p	−1, 52	0	contig_12464	2, 54	0	−12.00	PREDICTED: interleukin-20 receptor subunit beta-like [Salmo salar]
	ssa-miR-19c-3-5p	3, 04	0	contig_3735	−2, 11	0	−12.00	PREDICTED: myosin light chain kinase, smooth muscle-like isoform X3 [Salmo salar]
	ssa-miR-456-5p	0	8, 79	contig_63289	0	−3, 32	−12.00	PREDICTED: fibroblast growth factor 10-like [Salmo salar]
	ssa-miR-150-3p	0	−1, 6	contig_15517	0	2, 02	−12.10	ATPase inhibitor, mitochondrial precursor [Salmo salar]
	ssa-miR-19d-5p	0	−2, 68	contig_13064	0	2, 07	−12.00	ATP-binding cassette sub-family F member 2 [Salmo salar]
	ssa-miR-222b-5p	0	−1, 98	contig_78633	0	3, 86	−12.00	transposase [Salmo salar]
	ssa-miR-204-5p	−2, 52	0	contig_66355	2, 96	0	−12.00	transposase [Salmo salar]
	ssa-miR-18b-5p	0	1, 56	contig_19956	0	−17, 31	−12.00	PREDICTED: haptoglobin-like [Salmo salar]
	ssa-miR-456-5p	0	8, 79	contig_9077	0	−3, 2	−12.00	SPATA5 [Salmo salar]
	ssa-miR-456-5p	0	8, 79	contig_52147	0	−2, 72	−12.10	PREDICTED: MAPK/MAK/MRK overlapping kinase-like isoform X3 [Salmo salar]
	ssa-miR-456-5p	0	8, 79	contig_32835	0	−2, 64	−12.00	interleukin-17A/F3 [Salmo salar]
Intestine	ssa-miR-25-3-5p	4, 02	0	contig_19100	−2, 43	0	−30.60	PREDICTED: non-syndromic hearing impairment protein 5-like isoform X2 [Salmo salar]
	ssa-miR-30a-3-3p	0	2, 28	contig_58533	−3, 12	0	−30.50	PREDICTED: kynureninase-like [Salmo salar]
	ssa-miR-92b-3p	0	1, 22	contig_2303	2, 07	0	−30.40	PREDICTED: coronin-1B-like [Salmo salar]
	ssa-miR-214-5p	0	1, 35	contig_3322	2, 75	0	−30.10	PREDICTED: fibroblast growth factor receptor substrate 2-like [Salmo salar]
	ssa-miR-19c-5p	−2, 06	0	contig_51580	3, 29	0	−30.10	PREDICTED: protein-tyrosine kinase 6-like [Salmo salar]
	ssa-miR-92a-5p	3, 72	0	contig_4409	−4, 12	0	−30.00	PREDICTED: dihydropyrimidinase-related protein 4-like [Salmo salar]
	ssa-miR-125b-5p	0	1, 26	contig_4543	0	−1, 97	−29.80	PREDICTED: low-density lipoprotein receptor-related protein 2-like [Salmo salar]
	ssa-miR-30a-3-3p	0	2, 28	contig_3114	0	−1, 94	−29.00	PREDICTED: free fatty acid receptor 3-like [Salmo salar]
	ssa-miR-30a-3p	−1, 73	0	contig_44524	2, 17	0	−29.00	PREDICTED: annexin A2-like [Salmo salar]
	ssa-miR-92a-5p	3, 72	0	contig_51439	−3, 15	0	−28.90	PREDICTED: myosin heavy chain, fast skeletal muscle-like [Salmo salar]
	ssa-miR-15a-3p	1, 72	0	contig_9149	−2, 1	0	−28.40	PREDICTED: annexin A2-like [Salmo salar]
	ssa-miR-140-3p	−1, 53	0	contig_25304	4, 52	0	−28.40	PREDICTED: leucine-rich repeat-containing protein 58-like [Salmo salar]
	ssa-miR-456-3p	3, 35	0	contig_24874	−1, 73	0	−17.60	PREDICTED: sialic acid-binding Ig-like lectin 5 [Salmo salar]
	ssa-let-7i-5p	0	5, 51	contig_66659	0	−1, 37	−16.80	PREDICTED: dedicator of cytokinesis protein 3-like, partial [Salmo salar]
	ssa-let-7d-5p	4	0	contig_28976	−5, 05	0	−16.00	PREDICTED: fibroblast growth factor receptor substrate 2-like [Salmo salar]
Head kidney	ssa-miR-8157-3p	0	−2, 18	contig_122554	0	6, 77	−30.20	PREDICTED: E3 ubiquitin-protein ligase TRIP12-like isoform X1 [Salmo salar]
	ssa-miR-128-1-5p	−1, 57	0	contig_72217	6, 3	0	−29.60	SPATA5 [Salmo salar]
	ssa-miR-214-3p	0	1, 8	contig_21752	0	−1, 47	−29.50	PREDICTED: BTB/POZ domain-containing adapter for CUL3-mediated RhoA degradation protein 2 isoform X1 [Salmo salar]
	ssa-miR-8157-3p	0	−2, 18	contig_46746	0	1, 67	−29.50	PREDICTED: phospholipid-transporting ATPase 11C-like isoform X1 [Salmo salar]
	ssa-miR-205a-5p	0	1, 99	contig_45675	0	−1, 03	−29.40	PREDICTED: von Willebrand factor A domain-containing protein 7-like [Salmo salar]
	ssa-miR-214-3p	0	1, 8	contig_95191	0	−1, 84	−28.90	PREDICTED: fibrinogen beta chain-like [Salmo salar]
	ssa-miR-194a-3p	7, 34	0	contig_28663	−3, 31	0	−28.60	PREDICTED: laminin subunit alpha-4-like isoform X1 [Salmo salar]
	ssa-miR-462a-3p	−1, 86	0	contig_22955	2, 08	0	−34.50	PREDICTED: 3-keto-steroid reductase-like isoform X2 [Salmo salar]
	ssa-miR-122-5p	0	3, 15	contig_68369	0	−1, 89	−29.80	PREDICTED: transcription factor E2F7-like [Salmo salar]
	ssa-miR-125a-5p	0	2, 32	contig_31191	0	−1, 03	−27.70	PREDICTED: tectonic-2 [Salmo salar]
	ssa-miR-449a-5p	21, 47	0	contig_44769	−2, 24	0	−20.40	PREDICTED: guanine nucleotide-binding protein G(I)/G(S)/G(T) subunit beta-3-like isoform X1 [Salmo salar]
	ssa-miR-192b-5p	13, 39	0	contig_65873	−2, 18	0	−20.00	PREDICTED: E3 ubiquitin-protein ligase RNF144A-like [Salmo salar]
	ssa-miR-212a-5p	−1, 83	0	contig_49183	2, 56	0		PREDICTED: collagen EMF1-alpha-like [Salmo salar]

## Data Availability

Not applicable.

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
