# Peer review of "Whole-Genome Transcript Expression Profiling Reveals Novel Insights into Transposon Genes and Non-Coding RNAs during Atlantic Salmon Seawater Adaptation"

_biology, 2021, doi:10.3390/biology11010001_

Round 1
Reviewer 1 Report
This work is sound and interesting. I have only few minor comments.
Pag. 1 - Line 21. please specify what do you mean with whole-genome transcriptomic approach. I guess you meant whole transcriptome yet this definition should be better explained
Pag4 Line 153: change determinate with "determined" or alternatively modify the sentence to be more comprehensible.
Pag4 Line 153: here it is supposed that you are explaining the CGE index that is one of the main focuses of the manuscript. However, this is not clearly explained how it is computed and with which input data. Please clarify.
Pag6 Line 229: again I guess the term "determinate" is used instead of "determined"
Author Response
We do appreciate your suggestions to improve our manuscript. Please find below the responses to every question and/or comment that you have made. Furthermore, the corrections and required changes to the manuscript are highlighted in yellow.Pag. 1 - Line 21. please specify what do you mean with whole-genome transcriptomic approach. I guess you meant whole transcriptome yet this definition should be better explained. R Thanks for your suggestion, the term was changed and also the title was modified as “Whole-genome transcript expression profiling…” (line 21).
Pag4 Line 153: change determinate with "determined" or alternatively modify the sentence to be more comprehensible. R Thanks, was modified according to your suggestion (line 154).
Pag4 Line 153: here it is supposed that you are explaining the CGE index that is one of the main focuses of the manuscript. However, this is not clearly explained how it is computed and with which input data. Please clarify. R. Thanks for your comment. In the methods section was include more details of CGE index determination. (Lines 156-174)
Pag6 Line 229: again I guess the term "determinate" is used instead of "determined" R. Thanks for your comment. The term was changed to calculated (line 278)
Reviewer 2 Report
The authors used two pools with five individuals per sample point for library preparation. Subsequently, using RNA pools of each sampling point, double-stranded cDNA libraries were constructed. The same RNAs pools were used for small RNA libraries synthesis. Three biological replicates for RNA and smallRNAs were sequenced by the Hiseq (Illumina®, USA) platform. Generalized Linear Model (GLM) available in the CLC software was used for statistical analyses and to compare gene expression levels in terms of the log2 fold change (p = 0.005; FDR corrected).
Please explain better the experimental set-up and origin and number of three biological replicates (which are pools, in fact?) for the subsequent statistical analysis. At present it is difficult to judge the validity of the statistical analysis and therefore, the validity of the study.
Immune histochemistry analysis performed in Atlantic salmon exposed to GSC and SS conditions is said to show chloride cells migration, indicative of salmon adequate condition to SW transfer, in Figure S2. Furthermore, this condition is said to be confirmed by RT-qPCR analysis of ATPase- and ATPase- subunits (Figure S3).
The legends to Figs S1 and S2 are highly inadequate, please expand on the information to be read from the Figs.
Author Response
We do appreciate your suggestions to improve our manuscript. Please find below the responses to every question and/or comment that you have made. Furthermore, the corrections and required changes to the manuscript are highlighted light blue.The authors used two pools with five individuals per sample point for library preparation. Subsequently, using RNA pools of each sampling point, double-stranded cDNA libraries were constructed. The same RNAs pools were used for small RNA libraries synthesis. Three biological replicates for RNA and smallRNAs were sequenced by the Hiseq (Illumina®, USA) platform. Generalized Linear Model (GLM) available in the CLC software was used for statistical analyses and to compare gene expression levels in terms of the log2 fold change (p = 0.005; FDR corrected). Please explain better the experimental set-up and origin and number of three biological replicates (which are pools, in fact?) for the subsequent statistical analysis. At present it is difficult to judge the validity of the statistical analysis and therefore, the validity of the study.
R. Thanks for your comment. The RNA sequencing analysis was conducted using three libraries per sampling point and tissue (biological replicates; n=3). For each library preparation, five individuals from each experimental group were pooled. We clarified this point and the methods section was corrected. (Lines 140-147)
Immune histochemistry analysis performed in Atlantic salmon exposed to GSC and SS conditions is said to show chloride cells migration, indicative of salmon adequate condition to SW transfer, in Figure S2. Furthermore, this condition is said to be confirmed by RT-qPCR analysis of ATPase- and ATPase- subunits (Figure S3). The legends to Figs S1 and S2 are highly inadequate, please expand on the information to be read from the Figs.
R. Thanks for your comment. The legends of the supplementary figures were improved for better understanding (Please see supplemental material document). In specific, four supplementary figures were additionally added. The Figure S4 shows the global transcriptome modulation of Atlantic salmon exposed to GSC and SS conditions. The Figure S5 showed the differential expression analysis of transposable elements. The Figure S6 showed the RT-qPCR validation of candidate miRNAs and their putative target gene. Fold-changes values were calculated using Ssa-mir-455e5p as endogenous control. The Figure S7 displayed the correlation analysis of expression pairs among smoltification-related genes, TEs and differential expressed miRNAs.
Reviewer 3 Report
In this work, the authors used RNA-seq approach to determine expression changes of mRNAs and non-coding RNAs during smoltification of Atlantic salmon smolts. They focused their study on three tissues: gill, intestine and head kidney. Bioinformatic analyses seem to suggest that several mRNAs and miRNAs exhibit expression changes in response to salinity shock. When mapped to chromosome regions, the authors claim that differentially expressed genes are mostly related to transposable elements. The approach reported in this manuscript may be potentially interesting. However, this work is too preliminary and rather speculative. It falls short of providing a repertoire of smoltification-induced expression changes.
- My main concern is that most conclusions are not fully supported by experimental evidence. Although the authors mentioned a few differentially expressed mRNA and miRNA, how the genes are related to the smoltification process is not clear. The authors state that they identified a high abundance of transposable elements. However, this analysis is rather superficial, and the authors did not give more detail on these transposable elements. Moreover, the RNA-seq data were not validated by experimental evidence. Thus, the present study provides little insights into possible roles of transposable elements and non-coding RNAs in Atlantic salmon seawater adaptation.
- Most figures and tables presented in the manuscript will not be informative to a broad readership. Along with a very superficial description of the analyses, it is very difficult to understand how transcriptomic changes occur during smoltification or in response to salinity shock.
- It is disappointing that essentially the same sentences were used in the simple summary and in the abstract.
- Line 63, “Atlantic salmon transition to freshwater (FW) to seawater (SW)” should be Atlantic salmon transition from freshwater (FW) to seawater (SW)”.
- Figure S2, how RT-qPCR was performed? What is the internal control? It is unclear how the relative expression levels were normalized. There are also the same problems with figure S5. In addition, the authors should provide the exact gene names for transposase and ATPase inhibitor. They also need to provide primer sequences used for qPCR.
- Lines 240-241, “numerous transposase and transposable elements Tcb1, HSP70, and MCHII genes…”. What does this mean?
Author Response
We do appreciate your suggestions to improve our manuscript. Please find below the responses to every question and/or comment that you have made. Furthermore, the corrections and required changes to the manuscript are highlighted in green.
In this work, the authors used RNA-seq approach to determine expression changes of mRNAs and non-coding RNAs during smoltification of Atlantic salmon smolts. They focused their study on three tissues: gill, intestine and head kidney. Bioinformatic analyses seem to suggest that several mRNAs and miRNAs exhibit expression changes in response to salinity shock. When mapped to chromosome regions, the authors claim that differentially expressed genes are mostly related to transposable elements. The approach reported in this manuscript may be potentially interesting. However, this work is too preliminary and rather speculative. It falls short of providing a repertoire of smoltification-induced expression changes.
- My main concern is that most conclusions are not fully supported by experimental evidence. Although the authors mentioned a few differentially expressed mRNA and miRNA, how the genes are related to the smoltification process is not clear. The authors state that they identified a high abundance of transposable elements. However, this analysis is rather superficial, and the authors did not give more detail on these transposable elements. Moreover, the RNA-seq data were not validated by experimental evidence. Thus, the present study provides little insights into possible roles of transposable elements and non-coding RNAs in Atlantic salmon seawater adaptation. R. Thanks for your comment. We are agreeing that more clarification is required for the findings associated with the TEs. In this way we included a differential expression analysis of TEs in both experimental conditions, and all tissues to describe the transcriptome expression profiling. A differential expression analyses of TEs was include as supplementary results (Figure S5). Also, gene expression correlation analysis was included to evidencing the putative interplaying of miRNAs-mRNAs during the smoltification process. Here, smoltification-related genes and TEs were also included to calculate the correlation values (Figure 8). This information was included and discussed (Lines 224-296, 319-321, 342-43, 483-494, 532-536).
- Most figures and tables presented in the manuscript will not be informative to a broad readership. Along with a very superficial description of the analyses, it is very difficult to understand how transcriptomic changes occur during smoltification or in response to salinity shock. R. Thanks for your comment. However, according to our experience in similar transcriptome studies and the state-of-the-art to uncover molecular responses and mechanisms involved in relevant biological processes (e.g. fish smoltification), the study was properly designed and conducted. We understand that the complexity to disentangle the main molecular components associated with the parr-smolt transformation is not easy and require novel experimental and analytical approaches. For this reason, we proposed to explore how the whole-genome transcript expression profiling is modulated. Here, CGE analysis was proposed with the aim to unravel the transcriptional changes of fish exposed to different salinity conditions. The use of transcriptional thresholds to detect differentially expressed loci between mRNA-miRNA interactions allows the association with the chromosome architecture in Atlantic salmon. Using this approach, non-coding RNAs and transposable elements were uncovered, driving future research to explore how those genetic components are specifically involved in the regulation of smoltification-related genes. Interestingly, comparative CGE analysis among salmon species can be designed to understand how the evolution in anadromous fish is driven by mRNA-miRNA interplaying.
- It is disappointing that essentially the same sentences were used in the simple summary and in the abstract. R. Thanks for your comment. We the simple summary was modified. (Lines 17-29)
- Line 63, “Atlantic salmon transition to freshwater (FW) to seawater (SW)” should be Atlantic salmon transition from freshwater (FW) to seawater (SW)”. R. Thanks for your comment. The sentence was changed. (Line 63)
- Figure S2, how RT-qPCR was performed? What is the internal control? It is unclear how the relative expression levels were normalized. There are also the same problems with figure S5. In addition, the authors should provide the exact gene names for transposase and ATPase inhibitor. They also need to provide primer sequences used for qPCR. R. Thanks for your suggestion. In the methods was included the RT-qPCR protocol and the primer list was include as supplemental material table S4. (Lines 129-132; 230-262)
- Lines 240-241, “numerous transposase and transposable elements Tcb1, HSP70, and MCHII genes…”. What does this mean? R. Thanks for your comment. The sentence was re-writing for better understanding. (Lines 289-296)
Round 2
Reviewer 2 Report
The manuscript has been improved satisfactory, except for the legends of the supplementary figures. Again: please expand on the information in these legends such that the suppl figures + legends can be understood without direct reference to the text in the main manuscript
Author Response
Thanks for your comments. The legends of supplementary figures were improved as your suggestion. Please see the supplementary figures document.

Reviewer 3 Report
In my previous review, I feel that this work is too preliminary and rather speculative. In the revised manuscript, my main concerns were not addressed, and the results section showed little changes. The authors were even reluctant to experimentally validate their RNA-seq data by including more genes in the RT-qPCR analysis. Overall, it is unclear how transposon genes and non-coding RNAs change during smoltification.
Author Response
Thanks for your comments. We included a clustering profiling using k-means analysis to identify the TEs and miRNAs modulated during the smoltification. The transcription changes of TEs and miRNAs are visualized in four clusters between FW and GSC, and FW and SS, respectively. The findings revealed the modulation of these genetic elements during the seawater transfer, where tcb1 and tcb2 isoforms showed specific-expression patterns associated with the salinity condition (see supplementary figure 5 and 6, respectively). Here, we can suggest that the Atlantic salmon genome is actively transcribed through mobile elements and ncRNAs, evidencing their putative regulatory role in the fish biology. The experimental validation by RT-qPCR for selected TEs and miRNAs supports the gene expression changes associated with the salinity conditions. It is important to note that we previously reported the functional relationship between long-non coding RNAs (lncRNAs) and the smoltification in Atlantic salmon ([1]). This study highlighted the pivotal role of ncRNAs during the seawater transfer. Notably, the current study submitted for publication in Biology added another layer of scientific knowledge in anadromous fish species. Please see the supplementary figures document. (Lines 189-191; 303-310; 419-428)
[1] V. Valenzuela-Muñoz, J.A. Váldes, C. Gallardo-Escárate, Transcriptome Profiling of Long Non-coding RNAs During the Atlantic Salmon Smoltification Process, Marine Biotechnology 23(2) (2021) 308-320.
Round 3
Reviewer 3 Report
The authors performed RT-qPCR validation for some transposable elements and miRNAs. The data are presented in figures 5B and 6B. However, they need to provide more explanation of the analyses in the legends, for example what are T1, T2 and T3, and how did they normalize the mRNA relative expression.
Other points:
Lines 231 and 232, “considering as significant those correlations with P-value < 0.01”, however, in the legend to figure they describe that P-values > 0.01 were considered as significant. Please clarify this.
Line 620, I suppose that “GSA” should be “GSC”.
The manuscript needs thorough language editing or proofreading. Many sentences need rephrasing to be understood, just a few examples:
1. Lines 375, “positive regulation of genes expresses”.
2. Lines 651-652, “ The similar process were reported by Shwe, Østbye, Krasnov, Ramberg and Andreassen [19], that identified…”.
3. Line 653, “ the DE mRNAs genes”.
4. Lines 655-659, “ For instance, were identify BP…”.
Author Response
Dear reviewer 3,
please find the response to your comments. green highlight
The authors performed RT-qPCR validation for some transposable elements and miRNAs. The data are presented in figures 5B and 6B. However, they need to provide more explanation of the analyses in the legends, for example what are T1, T2 and T3, and how did they normalize the mRNA relative expression.
Thanks for your comment. The legend of figures S5B and S6B improved as you suggested. Furthermore, to better understand, the mean of T1, T2, T3 was explained and improved the figure S1, where explained the experimental design. Please see supplementary figures document.
Other points:
Lines 231 and 232, “considering as significant those correlations with P-value < 0.01”, however, in the legend to figure they describe that P-values > 0.01 were considered as significant. Please clarify this.
Thanks for your comment. The mistake was corrected in the manuscripts. (Line 550)
Line 620, I suppose that “GSA” should be “GSC”.
Thanks. The tipping mistake was corrected in the manuscripts. (Line 620)
The manuscript needs thorough language editing or proofreading. Many sentences need rephrasing to be understood, just a few examples:
Thanks for your suggestion. The manuscript was review for a native speaker
- Lines 375, “positive regulation of genes expresses”.
Thanks for your observation. The correct name of the GO term is “positive regulation of gene expression” and was corrected. (Line 375)
- Lines 651-652, “ The similar process were reported by Shwe, Østbye, Krasnov, Ramberg and Andreassen [19], that identified…”.
Thanks for your observation. The manuscript was review for a native speaker, and the sentence was corrected. (Lines 638-649)
- Line 653, “ the DE mRNAs genes”.
Thanks for your observation. The manuscript was review for a native speaker. (Lines 647-659)
- Lines 655-659, “ For instance, were identify BP…”.
Thanks for your observation. The manuscript was review for a native speaker. (Lines 647-659)